# Fabrication of Functional bioMOF-100 Prototype as Drug Delivery System for Breast Cancer Therapy

**DOI:** 10.3390/pharmaceutics14112458

**Published:** 2022-11-15

**Authors:** Renata Carolina Alves, Richard Perosa Fernandes, Renan Lira de Farias, Patricia Bento da Silva, Raquel Santos Faria, Christian Rafael Quijia, Regina Célia Galvão Frem, Ricardo Bentes Azevedo, Marlus Chorilli

**Affiliations:** 1Department of Drugs and Medicines, School of Pharmaceutical Sciences, São Paulo State University (UNESP), Rodovia Araraquara Jau, Km 01—s/n—Campos Ville, Araraquara 14800-903, Brazil; 2Department of Chemistry, Federal University of Mato Grosso (UFMT), Cuiabá 78060-900, Brazil; 3Departament of Chemical, Pontifícia Universidade Católica do Rio de Janeiro, Rio de Janeiro 22451-900, Brazil; 4Department of Genetics and Morphology, Institute of Biological Sciences, University of Brasilia (UnB), Campus Universitario Darcy Ribeiro—Asa Norte, Brasilia 70910-900, Brazil; 5Chemistry Institute, São Paulo State University (UNESP), Campus Araraquara, Araraquara 14800-060, Brazil

**Keywords:** cancer, delivery systems, bio-MOFs, curcumin, functionalization

## Abstract

Breast cancer is the most frequent cause of cancer death in women, representing the fifth leading cause of cancer death overall. Therefore, the growing search for the development of new treatments for breast cancer has been developed lately as well as drug delivery systems such as biocompatible metal–organic Frameworks (bio-MOFs). These may be promising and attractive for drug incorporation and release. The present study aims to develop a drug carrier system RCA (bioMOF-100 submitted to the activation process) containing incorporated curcumin (CCM), whose material surface is coated with folic acid molecules (FA) to promote the targeting of drug carrier systems to the tumor region. They were synthesized and characterized using several characterization techniques. The materials were submitted to drug encapsulation tests, whose encapsulation efficiency was 32.80% for CCM@RCA-1D. Using the ^1^H nuclear magnetic resonance (NMR) spectroscopy technique, it was possible to verify the appearance of signals referring to folic acid, suggesting success in the functionalization of these matrices. In vitro tests such as cell viability and type of cell death were evaluated in both series of compounds (CCM@RCA-1D, CCM@RCA-1D/FA) in breast tumor lines. The results revealed low toxicity of the materials and cell death by late apoptosis. Thus, these results indicate that the matrices studied can be promising carriers in the treatment of breast cancer.

## 1. Introduction

Cancer is one of the world’s killing diseases with an estimated 10 million deaths worldwide in 2019, according to the Global Health Data Exchange [1]. Breast cancer is one of the deadliest cancers, with reports of 700 million deaths per year and 20 million people with a prevalence [1]. The treatment of breast cancer generally involves the use of chemotherapy, radiotherapy, and surgery, with each type of breast cancer requiring a specific treatment [2].

Currently, several types of research are being carried out on the subject in an attempt to increase the treatment success rate, which is between 60–80% in the primary stage and 50% in metastasis [2,3]. Curcumin has been the target of different studies for the treatment of breast cancer, with 11 clinical trials registered at ClinicalTrials.gov.

Used as a nutraceutical, curcumin has been applied in popular medicine for centuries due to reports of pharmacological activities such as antitumorigenic, anti-inflammatory, antibiotic, antibacterial, anticancer, and antioxidant [4]. Curcumin is the majoritarian substance in the root of *Curcuma longa*, with the molecular formula C_21_H_20_O_6_ and a molecular mass of 368.28 g/mol [4]. Despite the potential in the pharmaceutical area reported by different researchers, with positive results against lung [5], colorectal [6], gastric [7], breast [8], cervical [9], vulva [10], ovary [11], and osteosarcoma [12], the substance exhibits limitations in its application due to its low aqueous solubility, which causes a low absorption in the gastrointestinal tract and results in low bioavailability [4].

Thus, strategies have been developed to try to improve the delivery of curcumin to specific sites to increase its bioavailability, such as drug delivery systems [13,14]. Among them, metal–organic frameworks (MOFs) are interesting due to their physicochemical stability and low toxicity, being already reported for the delivery of chemotherapeutic agents [15,16]. MOFs are made up of an extensive network of clusters or s metallic ions, coordinated to multidentate organic molecules [15,16].

Regarding the biological applications of this class of materials, biocompatible MOFs (bio MOFs) have been developed with bio elements (Zn, Fe, and Cu) and ligands such as amino acids, peptides, nitrogenous bases, etc. [17]. Bio-MOFs exhibit exceptional surface area and relatively large pore sizes (for drug encapsulation), intrinsic biodegradability (as a result of relatively labile metal–binder bonds), and the possibility of post-functionalization synthesis, which can improve the drug’s interaction with the pore walls [17]. These materials can also be scaled down to the nanometer scale to facilitate drug permeation into cells.

Seeking a specific interaction of the synthesized bio MOFs with breast cancer cells, it was functionalized with folic acid, which causes a greater selectivity of this carrier because tumor cells have overexpression of folate receptors (about 100–300 times more than in cells normal) via endocytosis [18,19,20].

In this context, we present the development of a drug carrier system RCA (bioMOF-100 submitted to the activation process) containing incorporated curcumin, whose material surface is coated with folic acid molecules to promote the targeting of these systems to the region tumor.

## 2. Materials and Methods

### 2.1. Materials and Physical Techniques

Unless otherwise mentioned, all chemicals were purchased from Sigma-Aldrich. Methanol, Dimethyl sulfoxide– d6 (DMSO-d6), Dimethyl sulfoxide (DMSO), N-hydroxysuccinimide (NHS), and 1-[3-(dimethylamino) propyl]-3-ethyl carbodiimide hydrochloride (EDC), Tween-20, N, N-Dimethylformamide (DMF) were purchased from Synth and CCM was purchased from Inlab Company. Powder X-ray diffraction (PXRD) patterns were collected using a Rigaku Rint 2000 powder diffractometer at 40 kV, 50 mA, for Cu Kα, (λ = 1.5406 Å) with a scan speed of 2°/min from 2 to 40° at a step size of 0.02°. Nuclear magnetic resonance spectroscopy of hydrogen (^1^H-NMR) (Bruker Advance III 600 MHz spectrometer). MOF samples were dried under N_2_ flow and digested in DMSO-d6 (1 mL) with deuterium chloride solution (35 wt.% in D_2_O, 10 µL). Fourier transform infrared spectroscopy (FTIR) experiments were performed on a Nicolet IS5 Thermo Scientific spectrometer. About 2 mg of RCA sample was ground and mixed with KBr. All spectra were analyzed from 400 to 4000 cm^−1^ with a resolution of 4 cm^−1^ and 16 scans. Optical microscope images of crystals were collected to verify their morphology. Scanning electron microscopy (SEM) images were obtained using a TOPCON SM-300 field emission gun (FEG) instrument using carbon as support. Simultaneous thermogravimetry and differential scanning calorimetry (TG-DSC) curves were obtained with a thermal analyzer Mettler Toledo TG/DSC1. A nitrogen atmosphere was used as a purge gas with a 50 mL/min flow rate. A heating rate of 10 °C/min was adopted running from room temperature to 1000 °C. Data were analyzed using the TA Universal Analysis software package. Elemental analyses of carbon, nitrogen, and hydrogen were performed with a Perkin Elmer 2400 Series II. Sorption isotherm studies of RCA were performed through an N_2_ atmosphere, using ASAP 2020-Micrometrics equipment.

### 2.2. Synthesis of RCA

The synthesis was performed according to the methodology described by An et al. [21]. The as-synthesized RCA was activated by applying a vacuum (50 mbar) at 160 °C for 24 h before further characterizations. After this process, the material was designated RCA-1.

### 2.3. Computational Methodology

#### 2.3.1. Analysis In Silico and Polymorphs

The 3D atomic coordinates of bioMOF-100 were acquired from the Cambridge Database (ID 833315), which were later converted into Cartesian coordinates using Mercury v2021 [22]. Then, through the polymorph generator module available in the CSD-materials tool pack, the possible conformation for the repeating unit is identified by the rotamer library, considering the geometric constraints of the input molecule. After selecting the best conformers, the geometric optimization was applied by the semi-empirical PM7 level with the MOPAC2016 [23,24]. We applied the root-mean-square deviation protocol for overlays into the Biovia Discovery Studio Visualizer4. Finally, for data analysis, calculation of voids, and generation of figures, we applied the CCDC Mercury.

#### 2.3.2. Molecular Docking

The structure of the DMF (N′, N′–dimethylformamide) ligand was constructed and previously minimized by molecular mechanics (MM) and semi-empirical PM7 levels in the Biovia DSV [25] and MOPAC2016, respectively. After that, the atomic coordinates were converted into PDB format. The structure of the macromolecule, resulting from the screening of polymorphs, was similarly converted. Then, both files were uploaded on the PachDock [26] online server, with the existing algorithm as default. Then, 100 poses were generated, but just 10 top-ranked scores were selected for a visual inspection. Finally, the high-resolution images were obtained from the Biovia DSV.

### 2.4. Incorporation of Curcumin into RCA

CCM (5 mg/mL) was mixed with 50 mg of RCA-1 in a 20 mL glass container. The mixture was shaken (200 rpm) for 3 days (CCM@RCA-1D and CCM@RCA-3D). The supernatant was collected after centrifugation (9000 rpm, 15 min). Materials were washed with methanol for 1 h to remove CCM in the solution [27]. To evaluate the presence of CCM in the CCM@RCA-1D and CCM@RCA-3D, the materials were subjected to FTIR, PXRD and thermogravimetric analysis (TGA), and differential scanning calorimetry (DSC) curves.

### 2.5. Estimation of Drug Loading

CCM quantification was determined by high-performance liquid chromatography (HPLC) according to the methodology reported by Alves et al. [4]. CCM calibration curves were prepared at concentrations between 0.5 and 75 µg/mL (Appendix A). Drug Loading Encapsulation Efficiency (DLE) was calculated using Equation (1):DLE (%) = (Wt/Wi) × 100%(1)
where Wt is the total amount of encapsulated CCM and Wi is the total amount of CCM added initially during the preparation.

### 2.6. Post-Synthetic Modification with Folic Acid

The conjugation of FA into CCM@RCA-1D was performed according to the methodology described by Chowdhuri et al. [28] with some modifications; an amount of 25 mg of FA was dissolved in 10 mL of DMSO and water 50:50 (*v*/*v*). The pH of the solution was maintained between 7.5–8.5 and then both 50 mg of EDC and NHS were added. Activation of the acid group of FA was carried out for 3 h at 30 °C. Then, 15 mg of CCM@RCA-1D was added to the activated FA solution. The reaction was continued for 12 h in the dark. Finally, the FA-targeted CCM@RCA-1D was washed several times and vacuum-dried. The post-synthetic modification was available by the ^1^H-NMR technique.

### 2.7. In Vitro Release Studies at pH-Controlled Conditions

For the in vitro study of the release, the profile was developed according to the study by Tiwari et al. [29]. The drug release was performed with pH stimuli in acidic (pH = 5) and physiological (pH = 7.4) conditions in Phosphate Buffered Saline (PBS) solution plus Tween-20 (0.5 *v*/*v*%). Initially, 2 mg of compounds CCM@RCA-1D and CCM@RCA-1D/FA were dissolved in 12 mL of PBS + Tween-20 solution under stirring at 500 rpm at 37 °C. After selected time intervals (0.5–132 h), the samples were centrifuged and a 1 mL aliquot of the supernatants was removed to determine the CCM content. Subsequently, 1 mL of PBS buffer + Tween-20 was added to the medium for replacement. The removed aliquots were then subjected to HPLC analysis to determine the amount of CCM released.

### 2.8. Cell Line and Culture Maintenance

In this study, three breast tumor cell lines were used: MCF-7 (ATCC^®^ HTB-22™) (Rio de Janeiro Cell Bank (RJCB), Rio de Janeiro, Brazil), MDA-MB-231(ATCC^®^ HTB-26™), and 4T1 (ATCC^®^ CRL-2539™) cell lines, purchased from American Type Culture Collection (ATCC, Manassas, VA, USA), as well as a murine embryonic fibroblast non-tumor cell line NIH/3T3 (ATCC^®^ CRL-1658™) (Rio de Janeiro Cell Bank (RJCB), Rio de Janeiro, Brazil). To carry out the experiments, aliquots of cells were thawed at room temperature and transferred to a cell culture bottle. The cells were kept at 37 °C in an atmosphere with 5% CO_2_ and 80% humidity. Cells were cultured in DMEM culture medium, supplemented with 10% (*v*:*v*) fetal bovine serum (FBS) and 1% (*v*:*v*) antibiotic solution (100 units/mL of penicillin and 100 mg/mL of streptomycin).

### 2.9. Cell Viability Assay

The inhibition of cell growth was measured by the colorimetric method of MTT (3-(4,5-Dimethylthiazol-2-yl)2,5-Diphenyl Tetrazolium Bromide). Cells measuring 5 × 10^3^ were plated in 96-well plates in the absence or presence of a compound in serial dilution (3.125–1000 µg/mL). For the dilutions, DMEM was used, supplemented with 0.4% DMSO and 0.025% Tween-80. After 24 h of treatment, the cells were incubated in an oven at 37 °C with an atmosphere containing 5% CO_2_. At the end of the incubation period, 15 μL of MTT at a concentration of 5 mg/mL were added to the cell culture wells, and after 3 h of incubation with MTT, 150 μL of DMSO was added. The quantification of optical density (OD) was measured in a spectrophotometer at 595 nm. The percentage of cell viability was determined from Equation (2):% Viability = Treatment Absorbance/Negative Control Absorbance × 100(2)

The IC_50_ value (concentration in μg/mL that inhibits 50% of cell viability) was determined using the dose-response curve using the GraphPad Prism 6.0 statistical program (GraphPad Software, San Diego, CA, USA).

### 2.10. Type of Cell Death Evaluated by Flow Cytometer

After treatment at the IC_50_ + SD concentrations of the respective samples, the cells were trypsinized and the cell pellet was resuspended in 500 µL of ice-cold PBS and centrifuged at 2000 rpm for 5 min at 4 °C. After discarding the supernatant, 100 µL of binding buffer (0.1 M Hepes (pH 7.4), 1.4 M NaCl and 25 mM CaCl_2_) was added to the pellet. Then, 5 µL of Annexin V solution (BD, USA) and 10 µL of PI solution (50 µg/mL) were added to each microtube. The microtubes were left for 15 min protected from light and at room temperature. Finally, 300 µL of binding buffer was added. The samples were placed on ice and taken for FACS analysis immediately. Hydrogen peroxide (H_2_O_2_) is an oxidizing agent that acts directly on the lipid membrane and DNA, leading to membrane rupture and causing cell death by necrosis. This, at a concentration of 29.4 µM, was used as a positive control for the experiment.

## 3. Results and Discussion

### 3.1. RCA Compound Characterization

The RCA compound was successfully synthesized. Table 1 and Figure 1 show vibrational spectrum bands in the infrared region (FTIR) similar to the bioMOF-100 reported by An et al. [21]; they are similar to the presence of both adenine ligands (ν_as_NH at 3335 cm^−1^ and ν_s_NH at 3184 cm^−1^) and BPDC (νC=O at 1665 cm^−1^ and νC–O at 1255 cm^−1^).

Moreover, the RCA has a different crystalline profile than that observed in the diffractogram of the bioMOF-100 compound (Figure 2). The RCA presents two first diffraction peaks at the same value of 2θ (4.98° and 5.70°) as the simulated material. In addition, there was a widening of these peaks and a decrease in their intensities, indicating that there was a change in the structure of the material that will be later confirmed with data from elemental analysis and molecular modeling. On the other hand, after the activation process (160 °C, 24 h, vacuum) the RCA-1 sample maintained the same diffraction pattern observed for the source compound, which is a very positive result since it reveals that the activation process did not change the crystalline nature of the compound.

From the analytical elemental analyses shown in Table 2, as well as the spectroscopic data, it was possible to propose the minimum formula C_16_H_17_N_3_O_6_Zn for the RCA compound. Comparing the results obtained with the literature data for the bioMOF-100 [21], it was possible to verify that there was a change in the original structure, a result that corroborates the X-ray powder diffraction data shown in Figure 2.

Moreover, the SEM-FEG images show that the RCA compound is formed by nanometric units of low crystallinity, which organize themselves to form particles with cubic morphology and size that varies from approximately 2 to 4 μm (Figure 3) (as evidenced by optical microscopy, see Appendix A). In addition, it has a high surface porosity (interparticle-type pores).

Figure 4 shows the variation of the T_onset_ of 374 → 355 °C and a loss of mass of 10.65% between 30–355 °C referring to the exit of the solvent molecules. This loss of 10.65% presented in the TGA curve of the sample RCA-1 is lower than the difference in mass of 15.27% observed in the thermogravimetric curve of the RCA compound (in the temperature range between 30 °C and T_onset_) (Figure 4), demonstrating that the activation process was efficient (but not complete) in removing the solvent molecules occluded in the porous structure of the material.

Finally, as illustrated in Figure 5, an isotherm with multilayer adsorption behavior was observed. The BET and Langmuir-specific areas of the RCA-1 compound are 5.32 and 7.90 m^2^/g, respectively. These values of specific areas can be explained by the fact that the structure contains solvent molecules adhered to the pores, as shown below in the computational method.

### 3.2. The Structural Proposition of RCA Using the Computational Method

#### 3.2.1. Polymorphs Prediction for the bioMOF-100

The low crystallinity of the RCA can be associated with the large unit cell and low crystalline density as suggested by internal channels [21]. Moreover, potential cavities may contain disordered solvents of low molecular density [30]. A crystalline material can present different crystal packing, giving rise to distinct crystalline phases bearing different physical–chemical properties [31]. Since there was a non-obvious correlation between the XRD data of RCA and those found for bioMOF-100 (CCDC 833315), we applied the in silico conformer generation prediction [22,32] to better understand the obtained coordination polymer. The bioMOF-100 exhibits vertices as octahedral zinc–adeninate building blocks linked by BPDC bridges [21]. The Cartesian coordinates of this subunit were adopted as input structures (Figure 6a). The truncated system of the SBUs does not allow rotations or variations in torsional angles. Similarly, restrictions were conducted for BPDC rings due to the resonance nature of aromatic bonds. Finally, all rotational links were considered free.

As a result, 67 different conformations were obtained whose overlaps are available in Figure 6b. The screening step considered knowledge-based algorithms (CSD library) with a scale that ranges 0–1 for the probability level of the best conformations. Next, a new screening was performed bearing in mind the RMSD for the heavy atoms between the generated structures and the reference molecule. As a result, the conformers 40, 14, 41, 12, and 19, which have higher probabilities, showed deviation results lower than 1 Å, Appendix A. Thus, we suggest that the crystal structure of the bioMOF-100 is already configured in its most stable form. The conformer 19 was selected for expansion of the coordination polymer, which exhibited the greatest deviation from the model structure, 0.9974 Å. As a result, a structure similar to bioMOF-100 was obtained (Figure 7), which has a cubic crystal system of high symmetry and spatial group 230 (Ia$\bar{3}$d). Channels appear along with the crystallographic directions [110] and [101]. Furthermore, along with the *c*-axis, the truncated structures of the SBUs connected by BPDC bridges are identified.

#### 3.2.2. Void Analysis

The low crystal density of the simulated structure is confirmed by the analysis of the voids, empty spaces capable of containing spherical probes of a given radius [33], Appendix A. Using a 1.2 Å radii probe, it was found that voids represent 85.5% of the unit cell. The same protocol was applied to the experimental structure of bioMOF-100, which presented a value of 75.3% of voids per cell. It is well known that a compact crystal with this same spatial group (Ia$\bar{3}$d) has approx. 25.407% voids per cell. Therefore, despite the single-phase characteristic of bioMOF-100, there is a possibility of reducing its crystallinity by increasing the vacancy of its pores [33]. This prediction can be directly linked to the RCA diffraction pattern, suggesting an explanation for the low-resolution signals.

#### 3.2.3. Host–Guest Molecular Docking

For docking simulation, a fragment of the modeled structure was considered, which displays at least eight available cavities with the entrance of the solvent channels along the crystallographic direction [110], Appendix A. From the implementation of the Patch Dock [26] search algorithm, docking calculations were performed out to identify the potential binding sites against a solvent molecule, DMF, previously minimized by the semi-empirical method PM7 [23]. Docking results for the best-scoring pose (Figure 8) reveal that the DMF molecule offers appropriate complementarity to the binding pocket of the simulated RCA structure, forming different recognition patterns. The DMF molecule mediates greater compaction between parallel BPDC subunits [31,32,33,34] and there are two non-classical hydrogen C―H···π interactions between DMF and the aromatic BPDC fragments. Moreover, there is a C―H···O contact from the methyl group of DMF and one of the O atoms of the BPDC carboxylate group. Finally, the distance between the C=O group of DMF and the centroid of the aromatic ring from adenine amounts to 3.56 Å which is greater at 0.34 Å than the sum of van der Waals radii for the respective subunits [34,35]. Therefore, it will not be considered an iteration between these last entities. Consequently, we hypothesized the relationship between the DMF removal by heating with the increasing degrees of freedom for BPDC ligands, resulting in the expansion of the channels in the structure of RCA and decreasing its crystallinity.

### 3.3. Characterizations of RCA Compound after Encapsulation Test

The RCA matrix was submitted to the drug CCM encapsulation test (17.35 Å size), allowing its entry into the pores of the material. In addition, the system can interact with the CCM through intermolecular interactions such as hydrogen bonds between the carbonyl of the carboxylate groups present in the BPDC bridge ligand of the material and the phenolic hydroxyl group of the CCM, aiding in the encapsulation.

Various tests were performed to evaluate the encapsulation efficiency (EE) of the drug in the matrix. For this, some parameters were tested, such as the relationship between the drug and the matrix, as well as the contact time, as shown in Table 3.

Two samples were selected for further characterization. Although the RCA compound has large vacancies with attached solvent molecules, its encapsulation efficiency was quite satisfactory. Based on the results obtained by this technique, it can be inferred that after drug encapsulation, the diffraction pattern of the CCM@RCA-1D material (incorporation of CCM with a contact time of 1 day) remained unchanged, indicating that the encapsulation process does not change the structure of the metal–organic matrix. Comparing the ^1^H NMR spectra of the samples after digestion with deuterated hydrochloric acid, it is observed that the RCA spectrum shows signals at 8.5 and 8.0 ppm referring to the adenine and BPDC ligands present in the structure of the material. By comparing the CCM@RCA-1D spectrum with the RCA and CCM spectra, it is possible to verify the presence of signals at 8.5 and 8.0 ppm referring to the ligands and also a set of signals between 7.75–6 ppm, thus demonstrating the presence of CCM in the matrix (Figure 9).

In addition, in the microscopy images, it was possible to verify that the cubic morphology of the material was maintained when the drug was incorporated with a contact time of 1 day (as shown in Figure 10B).

For the CCM@RCA-3D compound, it was found that there was a change in the profile of the diffractogram. Comparing the diffractogram of the CCM@RCA-3D compound with the diffractogram of the free CCM, it can be seen that the additional peaks that appear are from the free CCM, as shown in Figure 11. According to Vasconcelos et al. [36], this fact occurs when part of the drug is adsorbed on the surface of the material and not just incorporated into the pores. Moreover, there is no appearance of morphology (as shown in Figure 10C). This result evidences a possible rupture of the structure of the material; therefore, we did not continue with the studies with this compound due to the loss of its cubic morphology.

For the evaluation of the CCM encapsulation in the matrix, it was determined by HPLC (Appendix A). With the quantification of the drug, it was possible to determine the concentration of CCM contained in the supernatant of the RCA compound; the final concentration of the supernatant after 1 day of contact with the material was 67.2%. Thus, the CCM@RCA-1D compound presented an EE of 32.80%. Finally, for a better evaluation of the results obtained after the encapsulation test, based on Table 4, Appendix A, it can be inferred that there was a mass loss in the CCM@RCA-1D compound, which can be attributed not only to the exit of the solvent molecules but also to the beginning of the thermal decomposition of the compound drug from 190 °C.

### 3.4. Functionalization of the Compound with Folic Acid

Once the CCM molecules are occluded in the pores of the material, it is expected that the folic acid (FA) molecules are located on the surface of the matrix, as shown in Figure 12, so that they can be found by the folic acid receptors present in the tumor cells which allows a greater targeting of the carrier [37,38,39]. When comparing the spectrum of the material CCM@RCA-1D/FA with the spectra of RCA, CCM, and FA (Figure 9), the presence of signals at 8.5 and 8.0 ppm refer to the ligands, a set of signals at 7.75–6 ppm refer to CCM, and a set of signals in the regions 9.0–6.0 and 3.0–1.5 ppm are associated with FA, suggesting both the presence of CCM and FA in the matrix.

### 3.5. In Vitro Drug Release Assay

CCM@RCA-1D and CCM@RCA-1D/FA were subjected to in vitro drug release assays. The CCM release profile was studied in PBS + Tween-20 buffer (0.5 *v*/*v*%) at pH 5.0 and 7.4 at a temperature of 37 °C. Based on the equation of the line (Appendix A), it was possible to determine the amount of CCM released in the materials. In Figure 13 and Appendix A, it is observed that there was a greater release for the compounds without the presence of FA after 132 h of the experiment. Folic acid probably prevents the exit of CCM molecules since folate is present on the surface of the materials. Furthermore, it appears that for the CCM@RCA-1D sample at pH = 5.0, a higher release rate was obtained when compared with the test performed for the same sample at pH = 7.4. According to Zheng et al. [40], promoting a pH-sensitive release can decrease the premature release of the drug during blood circulation and increase its release due to the acidic microenvironments of the tumor regions, thus being beneficial for tumor inhibition in vivo.

### 3.6. Anticancer Activity

In Figure 14 it can be seen that the inhibition of cell proliferation of the compounds was different in the cell lines studied. For the non-tumor cell line, the precursors of the compounds (folic acid, adenine, and BPDC) and the compound CCM@RCA-1D/FA did not inhibit cell proliferation at the concentrations tested. On the other hand, CCM and the compounds RCA and CCM@RCA-1D were able to inhibit the proliferation of these cells with an IC_50_ of 25.87; 42.85, and 34.19 µg/mL, respectively (Figure 14 and Appendix A).

In the MCF-7 and 4T1 cell lines whose folic acid receptor expression is lower when compared with the MDA-MB-231 lineage, we found that the compounds CCM@RCA-1D and CCM@RCA-1D/FA were able to inhibit cell proliferation in both cells. However, when we analyzed the data for the MDA-MB-231 strain, whose folic acid receptor expression is higher, we verified that the functionalized compound (CCM@RCA-1D/FA) did not present a significant effect on cell proliferation at the concentration tested; it was not possible at first to evaluate the targeting action of folic acid in different breast tumor lineages. Moreover, RCA was the material that showed the greatest selectivity against the MCF-7 tumor line (IS = 0.76) (Appendix A).

After 24 h of exposure to CCM@RCA-1D and CCM@RCA-1D/FA samples, cells were labeled with annexin-V and PI and analyzed by flow cytometry. The adjustment of the parameters of the equipment for the adequate detection of the two markers was carried out using cells marked only with Annexin-V and cells marked only with PI after treatment with hydrogen peroxide (H_2_O_2_) for 24 h. Detection of Annexin-V labeling is represented by the X axis (FTIC-H) and IP detection corresponds to the Y axis (PerCP-H).

In MCF7 cells, CCM@RCA-1D treatment showed: 3.86% in Q1, 13.6% in Q2, 18.3% in Q3, and 64.2% in Q4. The treatment with CCM@RCA-1D/FA showed: 4.61% in Q1, 17.7% in Q2, 18.3% in Q3, and 59.4% in Q4. Similarly, in 4T1 cells, CCM@RCA-1D treatment showed: 14.5% in Q1, 46.6% in Q2, 3.71% in Q3, and 35.2% in Q4. The treatment with CCM@RCA-1D/FA showed: 7.95% in Q1, 28.0% in Q2, 4.09% in Q3, and 60.0% in Q4. These data suggest that in all three treatments both cells are in late apoptosis (Figure 15).

## 4. Conclusions

In the present work, it was possible to evaluate MOFs, especially bio-MOFs, as drug delivery systems with subsequent evaluation of their antitumor efficacy for breast cancer in vitro models.

Using solvothermal synthesis and starting from zinc (II) ions and two different ligands, adenine (N-donor) and biphenyldicarboxylate, BPDC (O-donor), and using the most diverse characterization techniques, we obtained a prototype of the bioMOF-100, called RCA in this thesis. Due to its porous characteristics, this material was evaluated in CCM drug encapsulation tests, whose incorporation efficiency obtained for the CCM@RCA-1D composite was 32.80%.

Using the ^1^H NMR technique, the presence of a set of signals referring to folic acid was verified, evidencing that the functionalization occurred effectively. The drug release assay showed that the compound CCM@RCA-1D/FA had a lower drug release rate than the compound CCM@RCA-1D. Furthermore, the release was more effective at acidic pH.

In in vitro studies, we used three breast tumor cell lines (MCF-7, 4T1, and MDA-MB-231) and a normal cell line (NIH/3T3) to evaluate the best cell viability of this series of compounds. The studies revealed very promising results in the MCF-7 and 4T1 strains. We also evaluated the cell death mechanism for the compounds CCM@RCA-1D and CCM@RCA-1D/FA in the two strains with the best cell viability results. In both strains (4T1 and MCF-7), the cells showed a mechanism of death by late apoptosis.

## Figures and Tables

**Figure 1 pharmaceutics-14-02458-f001:**
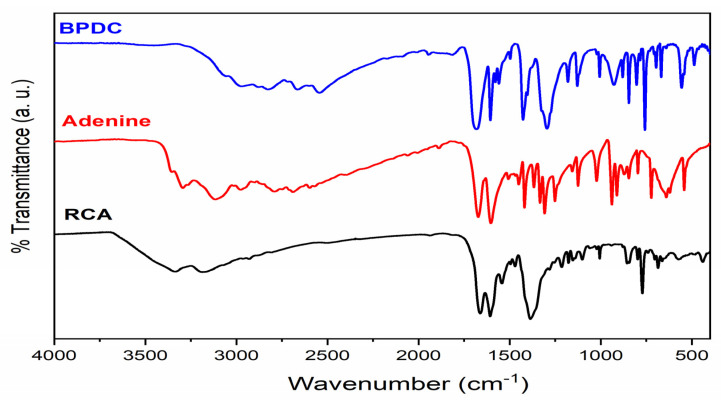
Infrared vibrational spectrum of the RCA compound.

**Figure 2 pharmaceutics-14-02458-f002:**
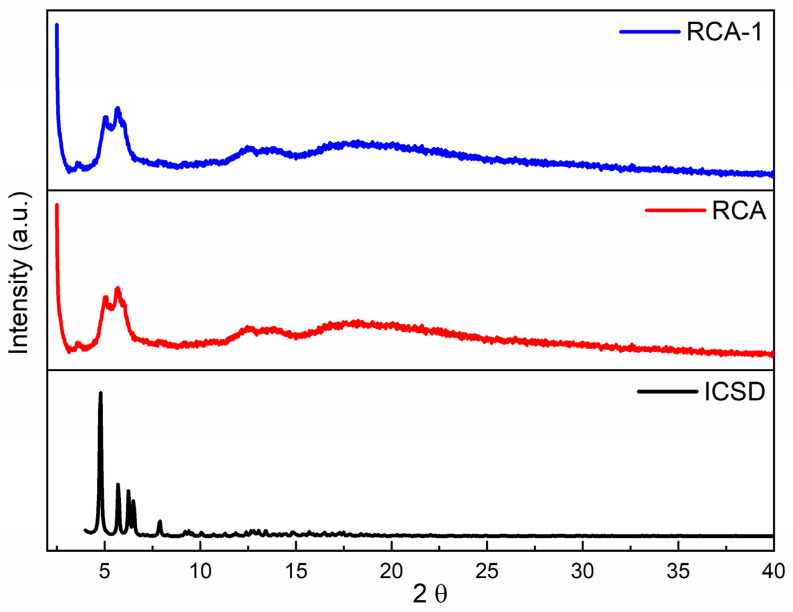
X-ray powder diffraction profiles for bioMOF-100: simulated by ICSD (file number: 833315), solid obtained as synthesized (RCA) and activated (RCA-1).

**Figure 3 pharmaceutics-14-02458-f003:**
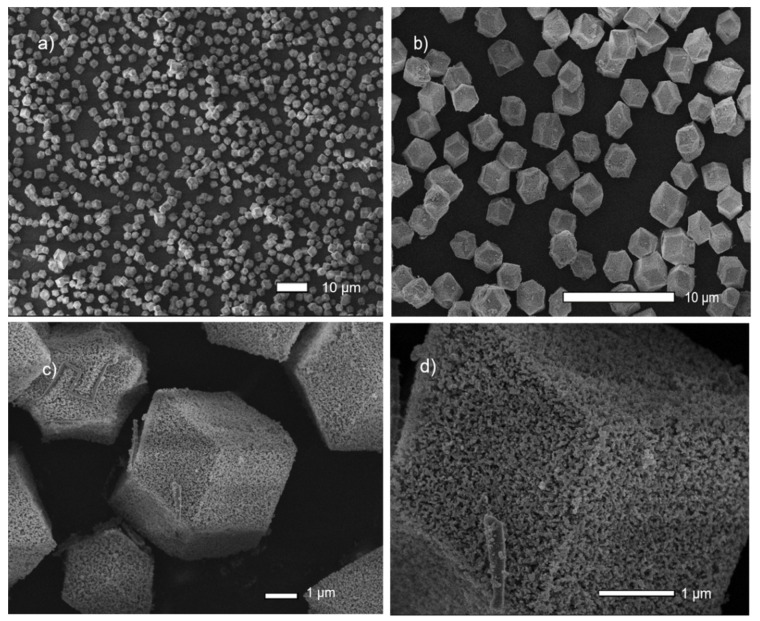
SEM-FEG images of the RCA compound obtained in this work. (**a**–**d**) are the composite of the RCA at different microscope magnifications.

**Figure 4 pharmaceutics-14-02458-f004:**
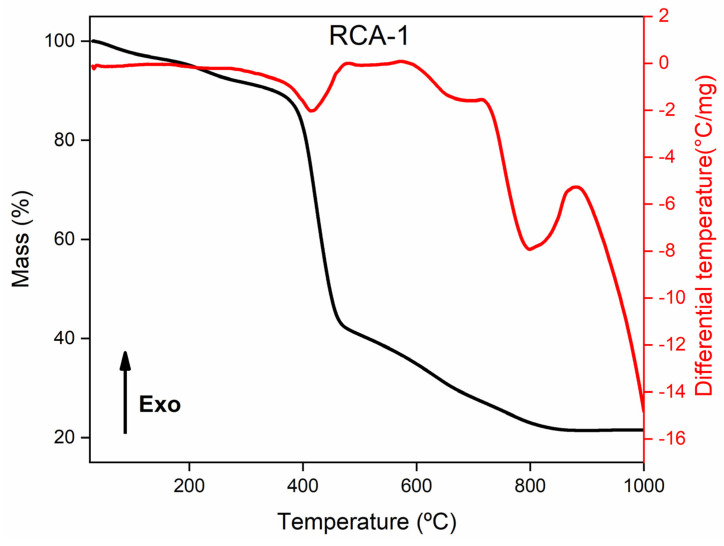
TGA−DSC curves obtained for the compound after the activation process (RCA−1 sample).

**Figure 5 pharmaceutics-14-02458-f005:**
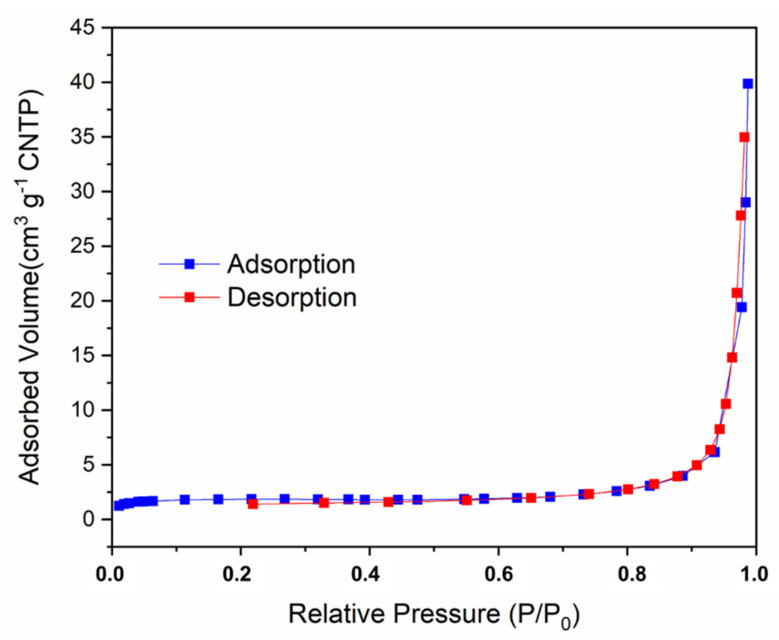
N_2_ physisorption isotherms for the activated compound (sample RCA−1).

**Figure 6 pharmaceutics-14-02458-f006:**
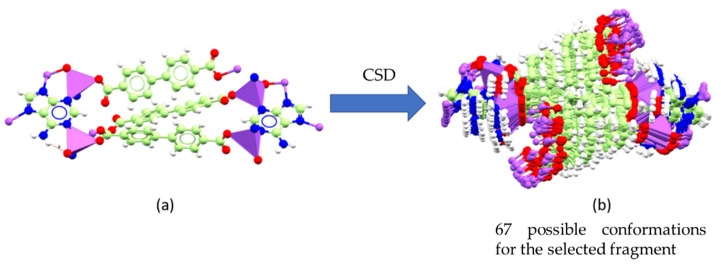
(**a**) BioMOF-100 repeating fragment using as a 3D input model for the generation of conformers. (**b**) Set of the 67 conformers generated from the replay subunit of bioMOF-100 (CCDC 833315). In the figure, Zn (II) ions are represented in purple color, N atoms in blue, C atoms in green, O atoms in red, and H atoms are blank/white.

**Figure 7 pharmaceutics-14-02458-f007:**
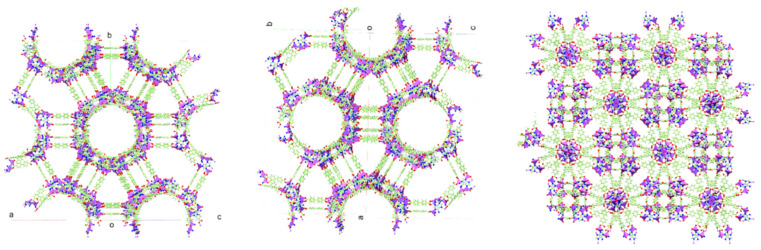
Expanded structure of the coordinating polymer from the conformer 19. View of the wide channels along with the directions [110] (**left**) and [101] **center**. The vision of the polymer along the crystallographic c-axis.

**Figure 8 pharmaceutics-14-02458-f008:**
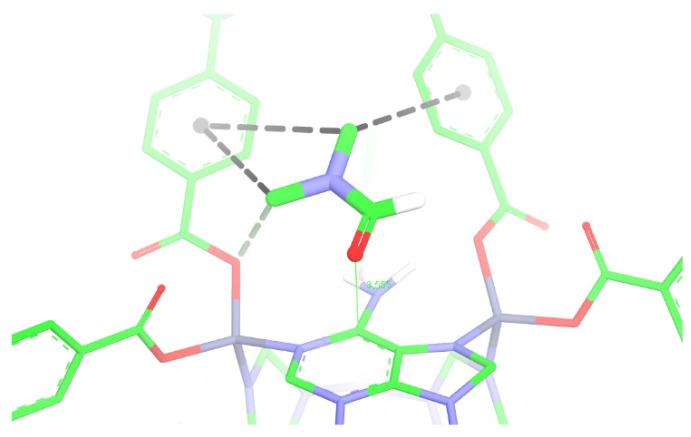
Docking poses for the best-scored solvent molecule hosted in one of the channels of the simulated structure for RCA. All non-polar H-atoms have been omitted for better visualization. In the pose, C-atoms are represented in green, N-atoms in blue, O-atoms in red, Zn (II)-ions in gray, and polar H-atoms are blank/white. Intermolecular contacts are represented by black dashed lines.

**Figure 9 pharmaceutics-14-02458-f009:**
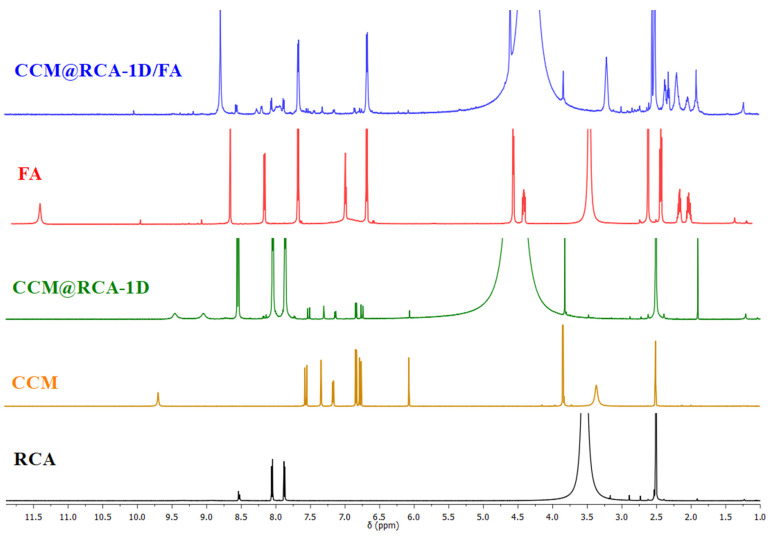
^1^H NMR spectra for compounds RCA, CCM, CCM@RCA-1D, FA, and CCM@RCA-1D/FA, in DMSO-d6.

**Figure 10 pharmaceutics-14-02458-f010:**
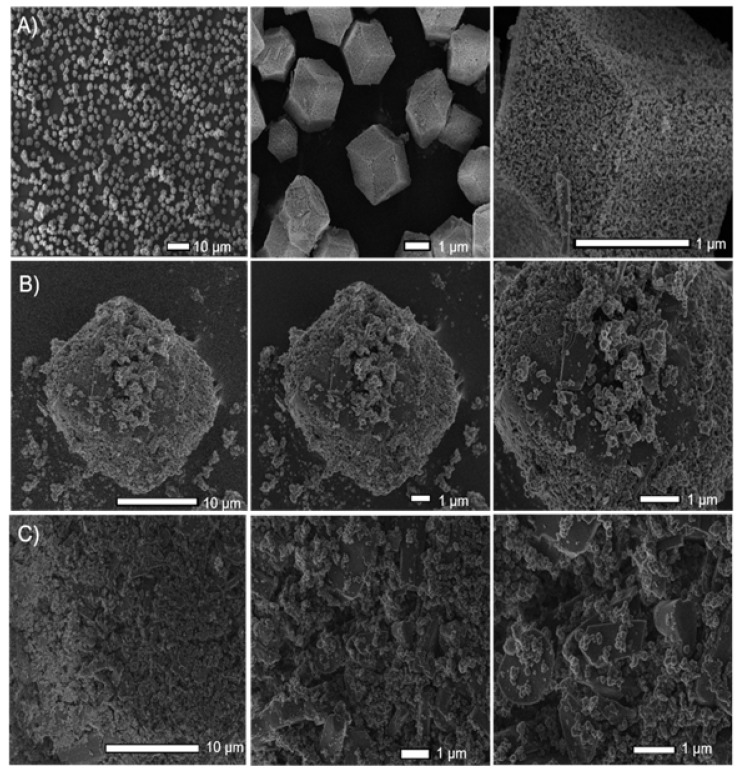
SEM-FEG images for RCA: (**A**) before CCM incorporation; (**B**) CCM@RCA-1D; and (**C**) CCM@RCA-3D.

**Figure 11 pharmaceutics-14-02458-f011:**
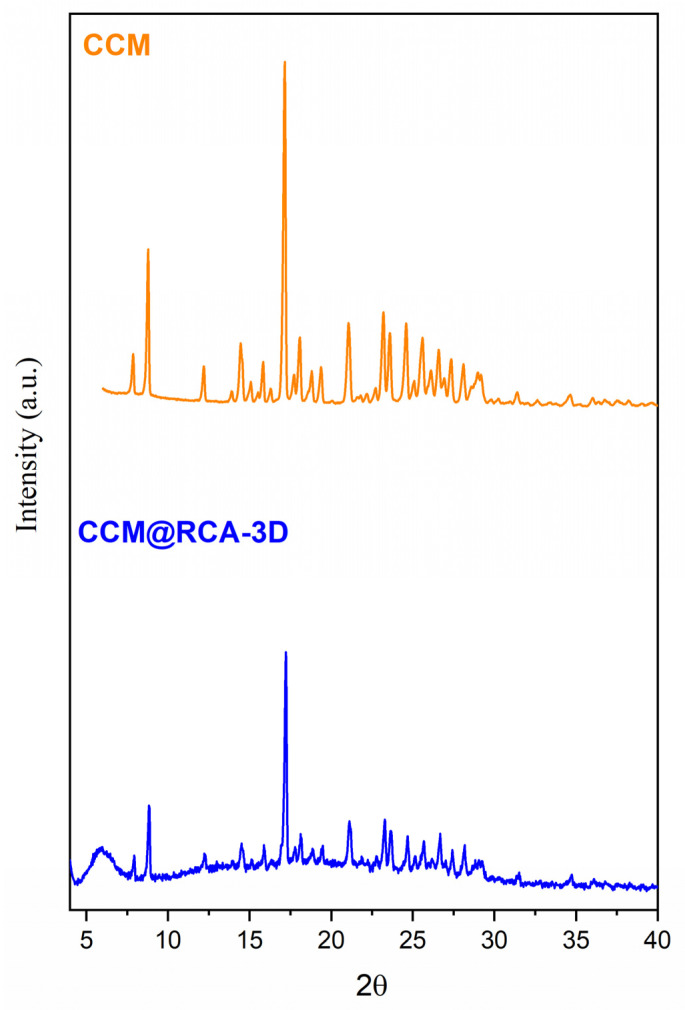
X-ray powder diffraction pattern of free CCM and CCM@RCA-3D sample.

**Figure 12 pharmaceutics-14-02458-f012:**
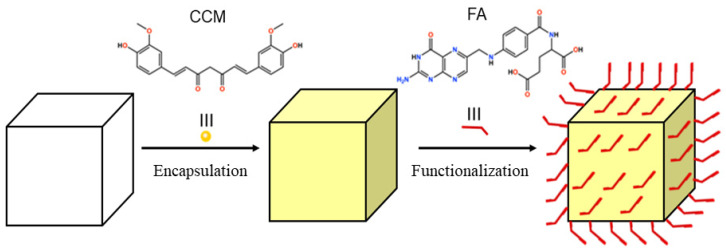
Representation of FA molecules on the surface of the matrix containing the drug.

**Figure 13 pharmaceutics-14-02458-f013:**
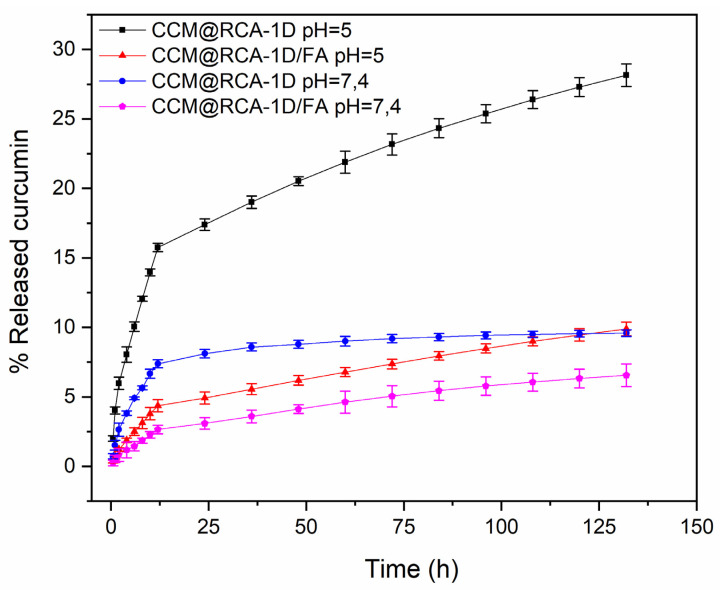
CCM release profiles in samples.

**Figure 14 pharmaceutics-14-02458-f014:**
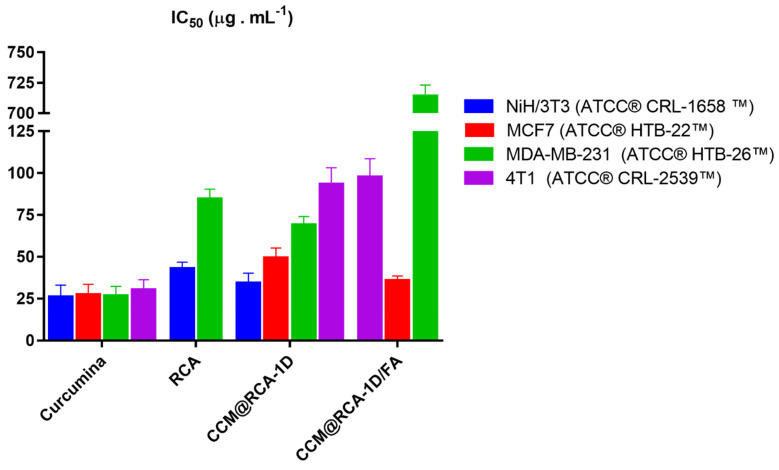
Graph representing IC_50_ values of RCA, CCM, compounds CCM@RCA−1D and CCM@RCA−1D/FA in cell line 4T1, MCF−7, MDA-MB-231, and NiH/3T3.

**Figure 15 pharmaceutics-14-02458-f015:**
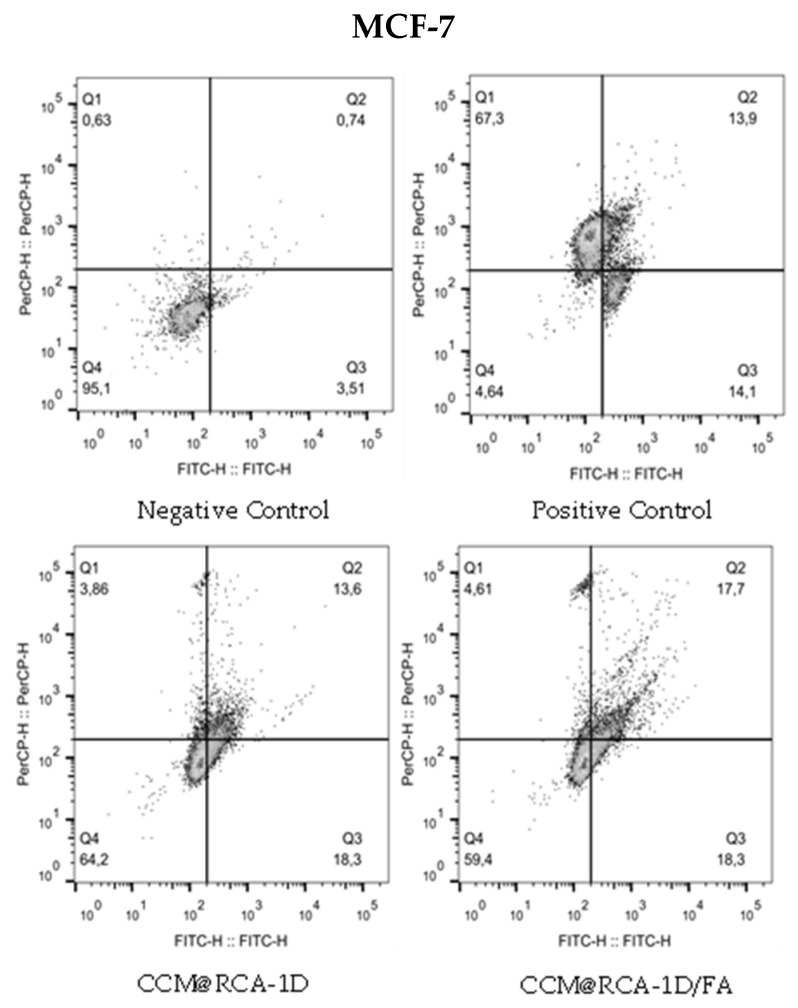
Graph representing the cell death mechanism of compounds CCM@RCA-1D and CCM@RCA-1D/FA in cell line 4T1 and MCF-7.

**Table 1 pharmaceutics-14-02458-t001:** Main frequencies in the infrared region of the material obtained (RCA) and that of the literature (bioMOF-100) [21].

Assignment	Wavenumber/cm^−1^	
	bioMOF-100	RCA
ν_as_NH	3341 br	3335 br
ν_s_NH	3185 br	3184 br
νCH_aliphatic_	2929 w	2927 w
νC=O + δNH_2_	1669 s	1665 s
νO-C=O + νC=N + νC=C	1607 s	1607 s
δC-NH	1547 w	1543 m
δ_np_OH + δN=CH	1467 w	1467 w
yC-H	1386 s	1384 s
νC-O	1255 w	1255 w
νC-NH_2_	1212 m	1215 m
δ_np_C_Ar_-H + δC-N=C + νC-N=C	1176 w	1171 w
δ_np_C_Ar_-H + δC-N=C + νC-N=C	1152 m	1153 w
δ_np_C_Ar_-H + δC-N-C	1097 m	1099 m
νNC + yNH_2_	856 m	857 m
νNC + yNH_2_	843 m	843 m
δ_fp_C_Ar_-C_Ar_-C_Ar_	773 s	774 s

ν = stretch; δ = angular strain; γ = out-of-plane strain; br = broad; m = medium; s = strong; w = weak; np = in plane; Ar = aromatic group; and fp = out of plane.

**Table 2 pharmaceutics-14-02458-t002:** Elemental analysis for RCA and comparison with bioMOF-100.

Results	% of Elements				
	C	H	N	O	Zn
RCA	46.78	4.08	11.90	21.57	15.67
bioMOF-100	45.43	7.51	14.84	24.59	7.63

**Table 3 pharmaceutics-14-02458-t003:** Tests to evaluate encapsulation efficiency.

Ratio RCA:CCM	Contact Time (Days)	EE (%)
1:1	1	32.80
1:3	1	32.75
1:3	2	38.82
1:3	3	44.00
1:3	4	41.06
1:3	7	17.66
1:4	4	12.19
1:4	7	23.65
1:2	7	25.13
2:1	7	23.67

**Table 4 pharmaceutics-14-02458-t004:** Thermogravimetric Analysis (TGA) of compounds RCA-1 and CCM@RCA-1D.

Compound	Attribution	Temperature (°C)	Weight Loss (%)
RCA-1	Solvents	T_amb_-55	10.65
	Elimination of organic matter	355–840	68.51
	Residue	840–1000	20.84
CCM@RCA-1D	Dehydration	T_am_-110	5.77
	Solvent + CCM	110–355	6.07
	Elimination of organic matter	355–790	63.06
	Residue	790–1000	24.71

## Data Availability

Not applicable.

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
