# Peer review of "Fabrication of Functional bioMOF-100 Prototype as Drug Delivery System for Breast Cancer Therapy"

_pharmaceutics, 2022, doi:10.3390/pharmaceutics14112458_

Round 1
Reviewer 1 Report
In this paper, authors developed a novel drug delivery system and potentially can be used to the treatment of breast cancer. Although the development process looks good, the provided in vitro functional data was poor and hard to support their conclusions.
Major points:
1) In Table 5, the authors should clearly plot the data and show the readers the effect of different compounds on cell proliferation, such as cell images or column/curve graphs, instead of the table.
2) In Figure 14, the authors did not show the indicates of x and y axis. Statistical results are in need.
3). The authors showed also look at the effect of different compounds on cell cycle.
4) The authors only evaluated the effect of compounds on 2D cultured cells, but the tumors are more complicated. In vivo tumor experiments are not necessary, but the authors should evaluated the effect of compounds on tumor spheres.
Author Response
We make changes to the text.
Point by point reply to the reviewers' comments of the manuscript: pharmaceutics-2016669 (“Fabrication of functional bioMOF-100 prototype as drug delivery system for breast cancer therapy.”)
We would like to thank again the reviewers for their constructive criticism and helpful suggestions. Below, please find our point-by-point replies to their comments, written in blue after each reviewer’s comments/suggestions.
In second new version of the manuscript, all recent modifications are highlighted in light blue.
1) In Table 5, the authors should clearly plot the data and show the readers the effect of different compounds on cell proliferation, such as cell images or column/curve graphs, instead of the table.
We added a figure showing the different cytotoxic effects of the compounds (Line 446) and the table was added in supplementary material
2) In Figure 14, the authors did not show the indicates of x and y axis. Statistical results are in need.
We also add this information to the result, indicating that the X axis corresponds to annexin and the Y axis corresponds to propidium iodide. The statistical data are in the figure in each of the quadrants, in addition, we made a description in the article (line 454-466).
3). The authors showed also look at the effect of different compounds on cell cycle.
For the moment, we only want to show these results in vitro, since they are promising for in vivo studies. But we are going to take into account for the future publication that they will be made in the following years.
4) The authors only evaluated the effect of compounds on 2D cultured cells, but the tumors are more complicated. In vivo tumor experiments are not necessary, but the authors should evaluated the effect of compounds on tumor spheres.
We previously carried out this study before it was sent to the scientific journal, unfortunately some microparticles precipitated and others did not, therefore, the cytotoxic effects did not reach a conclusion on this study.
Reviewer 2 Report
In the manuscript "Fabrication of functional bioMOF-100 prototype as drug delivery system for breast cancer therapy", the authors prepared the functional bioMOF-100 prototype o to develop a drug carrier system RCA containing incorporated curcumin (CCM), whose surface of these materials was coated with folic acid molecules (FA), to promote the targeting of these systems to the tumor region. Some of the results are sound, and the discussion is clear in some aspects. In conclusion, overall manuscript results and discussions are good. It would be of interest to the readership of this journal. Therefore, I would recommend publishing it after minor revising the following.
1. XRD spectra of RCA or RCA-1 are not clear. How about the sample quantity? If possible, replace the excellent quality XRD spectrum.
2. Do you have any direct evidence of elemental analysis? Why not EDS analysis or elemental mapping? If so, the results will be more evident and promising.
3. Drug loading efficiency, release profile, and other biological evolutions are good.
Author Response
We make changes to the text.
Point by point reply to the reviewers' comments of the manuscript: pharmaceutics-2016669 (“Fabrication of functional bioMOF-100 prototype as drug delivery system for breast cancer therapy.”)
We would like to thank again the reviewers for their constructive criticism and helpful suggestions. Below, please find our point-by-point replies to their comments, written in blue after each reviewer’s comments/suggestions.
In second new version of the manuscript, all recent modifications are highlighted in light blue.
- XRD spectra of RCA or RCA-1 are not clear. How about the sample quantity? If possible, replace the excellent quality XRD spectrum.
We modify the image to obtain a clear result. (Line 232)
- Do you have any direct evidence of elemental analysis? Why not EDS analysis or elemental mapping? If so, the results will be more evident and promising.
We appreciate your advice. We believe that these data are evident with respect to the literature, by comparing the minimum formula of bio-MOF-100.
- Drug loading efficiency, release profile, and other biological evolutions are good.
We appreciate your opinion.
Reviewer 3 Report
The manuscript provides an extensive focus on Fabrication of functional bioMOF-100 prototype as drug delivery system in the context of targeting breast cancer. The work is well planned, reasonably written, literature review is up-to date and well discussed, and the data are conceiving with the aspects of developed biofunctionalized MOF technology.
However, the article won’t be suitable to attract high attention with the readers of the Pharmaceutics due to poor introduction section and unavailability of schematic diagram. As such, I recommend acceptance after considering the suggestions as described above and below:
Ø I suggest authors to look carefully in line 133 and figure 4 which signifies the study is to be known as Thermo Gravimetric Analysis (TGA) not as TG-DSC curve. Thus, I suggest authors to look carefully to modify it
Ø As authors have used EDC-NHS coupling protocol for conjugation of FA with MOF. However, I am a highly interested to know how many molecules of FA was finally coupled with each MoF in final product.
Ø In vitro drug-release studies are not convincing in my opinion. The authors should also check the drug release in more stressful environments, such as FBS 50%, that better mimic the complexity of blood.
Ø Hemolytic study needs to perform to signify the biocompatibility of MOF and its other functionalized systems.
Author Response
We make changes to the text.
Point by point reply to the reviewers' comments of the manuscript: pharmaceutics-2016669 (“Fabrication of functional bioMOF-100 prototype as drug delivery system for breast cancer therapy.”)
We would like to thank again the reviewers for their constructive criticism and helpful suggestions. Below, please find our point-by-point replies to their comments, written in blue after each reviewer’s comments/suggestions.
In second new version of the manuscript, all recent modifications are highlighted in light blue.
Ø I suggest authors to look carefully in line 133 and figure 4 which signifies the study is to be known as Thermo Gravimetric Analysis (TGA) not as TG-DSC curve. Thus, I suggest authors to look carefully to modify it
We appreciate your suggestion; we modify this typographical error throughout the article.
Ø As authors have used EDC-NHS coupling protocol for conjugation of FA with MOF. However, I am a highly interested to know how many molecules of FA was finally coupled with each MoF in final product.
We carry out several studies such as UV-VIS; IR; HPLC to determine the amount of FA that was incorporated into the MOFs, however, we did not obtain clear results. For that reason, one of the ways to check FA conjugation was by means of NMR and it was discussed in the manuscript (line 410-413).
Ø In vitro drug-release studies are not convincing in my opinion. The authors should also check the drug release in more stressful environments, such as FBS 50%, that better mimic the complexity of blood.
For the moment, we only want to show these results in vitro, since they are promising for in vivo studies. But we are going to take into account for the future publication that they will be made in the following years. On the other hand, the analysis of curcumin release in FBS, we must perform other more extensive methods for its quantification.
Ø Hemolytic study needs to perform to signify the biocompatibility of MOF and its other functionalized systems.
For the moment, we only want to show these results in vitro, since they are promising for in vivo studies. But we are going to take into account for the future publication that they will be made in the following years.
Round 2
Reviewer 1 Report
Revised version is good to me now.